People with gambling disorder and risky alcohol habits benefit more from motivational interviewing than from cognitive behavioral group therapy

Josephson Henrik 1 henrik.josephson@ki.se
Carlbring Per 2
Forsberg Lars 1
Rosendahl Ingvar 1
1 Centre for Psychiatry Research, Department of Clinical Neuroscience, Karolinska Institute & Stockholm Health Care Services, Stockholm County Council , Stockholm , Sweden
2 Department of Psychology, Stockholm University , Stockholm , Sweden
Patton Bob
Electronic publication date: 2016 Mar 31
Publication date: 2016
Volume: 4
Electronic Location ID: e1899
Received 2015 Nov 17; Accepted 2016 Mar 15
Copyright: ©2016 Josephson et al.
Copyright year: 2016
Copyright holder: Josephson et al.
License: This is an open access article distributed under the terms of the Creative Commons Attribution License, which permits unrestricted use, distribution, reproduction and adaptation in any medium and for any purpose provided that it is properly attributed. For attribution, the original author(s), title, publication source (PeerJ) and either DOI or URL of the article must be cited.
License URL: https://creativecommons.org/licenses/by/4.0/

Keywords: Gambling, Alcohol problem, Psychological treatment, Comorbidity

Funding: Swedish National Institute of Public Health This study was funded by a grant from the Swedish National Institute of Public Health. The funders had no role in study design, data collection and analysis, decision to publish, or preparation of the manuscript.

==============================
Background. Effective psychological treatment, including cognitive behavioral therapy and motivational interviewing (MI), is available for people with problematic gambling behaviors. To advance the development of treatment for gambling disorder, it is critical to further investigate how comorbidity impacts different types of treatments. The purpose of this study was to investigate whether screening for risky alcohol habits can provide guidance on whether people with gambling disorder should be recommended cognitive behavioral group therapy (CBGT) or MI.

Methods. The present study is a secondary analysis of a previous randomized controlled trial that compared the effects of CBGT, MI and a waitlist control group in the treatment of disordered gambling. Assessment and treatment was conducted at an outpatient dependency clinic in Stockholm, Sweden, where 53 trial participants with gambling disorder began treatment. A modified version of the National Opinion Research Centre DSM-IV Screen for gambling problems was used to assess gambling disorder. The Alcohol Use Disorders Identification Test (AUDIT) was used to screen for risky alcohol habits.

Results. The interaction between treatment and alcohol habits was significant and suggests that patients with gambling disorder and risky alcohol habits were better helped by MI, while those without risky alcohol habits were better helped by CBGT.

Conclusions. The results support a screening procedure including the AUDIT prior to starting treatment for gambling disorder because the result of the screening can provide guidance in the choice of treatment. Patients with gambling disorder and risky alcohol habits are likely to be best helped if they are referred to MI, while those without risky alcohol habits are likely to be best helped if they are referred to CBGT.

Introduction

Worldwide, .3%–5.3% of adults suffer from gambling problems (Wardle et al., 2011). Disordered gambling is a diagnosis described in the Diagnostic and Statistical Manual of Mental Disorders, 5th edition (DSM-5; American Psychiatric Association, 2013), as a persistent and recurrent problematic gambling behavior leading to clinically significant impairment or distress. The diagnosis shares several characteristics with substance-related disorders. Common features include preoccupation, increased tolerance, loss of control, withdrawal symptoms, and family and job disruption (American Psychiatric Association, 2013).

Meta-analyses and systematic reviews have provided evidence for the efficacy of psychological treatment for gambling disorder (Gooding & Tarrier, 2009; Hodgins, Stea & Grant, 2011; Yakovenko et al., 2015). Results from a meta-analysis revealed that various forms of cognitive behavioral therapy (CBT) and motivational interviewing (MI) showed large and significant effect sizes in the 0–3-months period post treatment, with enduring effects at the 24-month (or later) follow-up (Gooding & Tarrier, 2009). Effect sizes were highly significant despite variability in the populations being treated, severity of gambling problem, and type of gambling (Gooding & Tarrier, 2009).

It is well known that gambling disorder is highly comorbid with other psychiatric disorders (Bischof et al., 2013; Lorains, Cowlishaw & Thomas, 2011; Petry, Stinson & Grant, 2005). Data derived from a large national sample in the United States indicate that the most frequently reported lifetime comorbid condition among people with gambling disorder was alcohol use disorder 73.2% (Petry, Stinson & Grant, 2005); the corresponding figure in a large German study was 61.7% (Bischof et al., 2013). What is even more interesting from a clinical perspective is that the results of a recent review on co-morbidity among gamblers seeking treatment for their gambling problems point in the same direction, with rates of current alcohol use disorders at 21.2% (Dowling et al., 2015). Gamblers’ alcohol consumption while gambling and the effect of alcohol on their gambling behavior are of particular interest. Most regular video lottery terminal (VLT) gamblers (73%) said that they prefer to drink alcoholic beverages while gambling (Stewart et al., 2002), and up to 80% of gamblers without a gambling disorder diagnosis reported a consumption of four to ten alcoholic drinks during their last gambling session (Baron & Dickerson, 1999). In other words, gamblers often drink while gambling, and it has been shown that those who do tend to engage in more risky gambling behaviors (Cronce & Corbin, 2010; Ledgerwood et al., 2009). People with gambling disorder and co-occurring alcohol use disorders reported greater levels of problematic gambling (Welte et al., 2004) and were more likely to have psychiatric comorbidity than those without alcohol use disorders (Abdollahnejad, Delfabbro & Denson, 2014). Gamblers with alcohol problems are also at an increased risk of relapse after quitting gambling (Hodgins & El-Guebaly, 2010).

In a study that mapped the drinking patterns of people with gambling disorder, entry into gambling treatment was temporally associated with reduction in alcohol use, but gamblers with risky alcohol habits were still less likely to adhere to gambling treatment (Rash, Weinstock & Petry, 2011). One study suggested that alcohol problems were linked to poor compliance in individual CBT treatment for gambling disorder (Milton et al., 2002). The study reported that people with gambling disorder comorbid with alcohol problems were 2.5 times more likely to drop out of treatment than people with gambling disorder without alcohol problems (Milton et al., 2002). This result has failed to be replicated in subsequent research on individual CBT (Leblond, Ladouceur & Blaszczynski, 2003) and multimodal CBT (Stinchfield, Kushner & Winters, 2005). Reports on the relationship between alcohol problems and dropout in the treatment of gambling disorder are of clinical interest, but there is no research on how co-morbid conditions such as alcohol problems affect the outcome of patients who actually pursue and remain in treatment for gambling disorder. Neither is there any research on whether different treatment forms, such as CBT and MI, differ in sensitivity to co-occurring alcohol problems. To advance the development of treatment for gambling disorder, it is critical to investigate further how comorbidity impacts different types of treatments for gambling disorder (Dowling et al., 2015; Hodgins, Stea & Grant, 2011; Petry, Stinson & Grant, 2005).

Because an alcohol use disorder is the most common comorbid condition among people with gambling disorder (Bischof et al., 2013; Dowling et al., 2015; Petry, Stinson & Grant, 2005) and contributes to a loss of control over gambling (Cronce & Corbin, 2010; Ledgerwood et al., 2009), more severe gambling problems (Welte et al., 2004), higher rates of psychiatric comorbidity (Abdollahnejad, Delfabbro & Denson, 2014), impaired adherence to gambling treatment (Milton et al., 2002; Rash, Weinstock & Petry, 2011), and an increased likelihood of gambling (Hodgins & El-Guebaly, 2010), it is of great clinical interest to see whether the condition has different affects on the outcome of widely used therapies, such as cognitive behavioral group therapy (CBGT) and MI. The purpose of this study was to investigate whether screening for risky alcohol habits can provide guidance on whether people with gambling disorder should be recommended CBGT or MI. The analysis in the present study is based on a data set from a previous randomized controlled trial that compared the effects of CBGT, MI, and a waitlist control in the treatment of gambling disorder (Carlbring et al., 2010). At baseline the three randomized groups had no significant differences in gambling-related measures, levels of depression, or alcohol consumption. In the trial the CBGT and MI groups showed greater reductions in the symptoms of gambling disorder than the no-treatment control group. Both CBGT and MI generated significant within-group decreases on gambling-related outcome measures up to the 12-month follow-up. However, no differences in outcome measures were found between CBGT and MI at any point (Carlbring et al., 2010).

Methods

Design

The present study is a secondary analysis of a previous randomized controlled trial that compared the effects of CBGT, MI, and a waitlist control in the treatment of gambling disorder (Carlbring et al., 2010). The study was approved by the regional ethics committee in Stockholm (2005/5:5), and informed written consent was collected from each participant.

Recruitment and participants

Between June 2005 and December 2006, 80 people with gambling disorder began treatment at an outpatient dependency clinic in Stockholm, Sweden. A total of 53 trial participants were included in the present study. Reasons for exclusion were not providing baseline data (n = 2) and not providing data at the 6-month treatment follow-up (n = 25). Participants received two cinema tickets for participating in the treatment follow-up.

Diagnostic and data collection procedures

Prior to starting treatment, all participants went through a 60- to 90-minute in-person interview conducted by a clinical psychologist trained in the assessment procedures. The interview included demographic questions and a set of self-report measures, including the National Opinion Research Centre DSM-IV Screen for gambling problems (NODS; Gerstein et al., 1999), the Timeline Followback (TLFB) calendar (Weinstock, Whelan & Meyers, 2004) and the Alcohol Use Disorders Identification Test (AUDIT; Babor et al., 1989). The participants were asked to fill out the set of self-report measures again at the 6-month follow-up.

Measures

The NODS (Gerstein et al., 1999), modified to assess gambling at one month instead of one year, was used to assess gambling disorder. The use of the 1-month version of the instrument has not seemed to affect the instrument’s reliability or validity. A comparison of the internal consistency between the NODS lifetime, past-year. and 3-month versions has shown Cronbach’s alphas of .86, .87, and .87 respectively (Wulfert et al., 2005). The total score, ranging from 0 to 10, is normally used to identify pathological gambling (scores 5 and above) according to DSM-IV (American Psychiatric Association, 2000). The instrument was modified to assess gambling disorder according to DSM-5 by eliminating the illegal acts criterion and lowering the threshold for diagnosis to 4 criteria of a possible 9. Recent research indicates that the increased sensitivity of the DSM-5 gambling disorder diagnosis successfully identifies a broader group of gamblers with clinically significant gambling-related problems (Rennert et al., 2014). Participants included in the present study were those assigned with NODS scores of 4 through 9 at baseline. A TLFB calendar (Weinstock, Whelan & Meyers, 2004) was used to assess the number of days gambled in the last 30 days.

The AUDIT (Babor et al., 1989) was used to assess risky alcohol habits. The instrument is a 10-item multiple-choice self-report inventory with a total score ranging from 0 to 40. Scores of 0–7 for men and 0–5 for women (Zone 1) indicate low-risk drinking. Scores of 8–15 for men and 6–13 for women (Zone 2) indicate hazardous and harmful alcohol use. Scores of 16–19 for men and 14–17 for women (Zone III) indicate a medium level of alcohol problems with a probable alcohol-related diagnosis. Finally, scores above 19 for men and 17 for women (Zone IV) indicate a high level of alcohol problems, with a probable alcohol-related diagnosis. The AUDIT accurately assesses the severity of problematic alcohol use behaviors across a wide range of contexts and populations at risk (Allen et al., 1997). When administered as part of a larger battery of tests in a primary care setting, the AUDIT showed test–retest reliability after a 6-week interval with a correlation of r = .88 and an internal consistency reliability of α = .85 (Daeppen et al., 2000). In the present study AUDIT scores were analyzed in two ways: first with Zones II–IV as three separate categories vs. Zone I (reference) and then as a dichotomized factor with Zones II–IV combined vs. Zone I.

Treatments

The CBGT treatment (n = 25) was administered in closed groups conducted as one 3-hour session per week for 8 weeks. The treatment was manualized (Ortiz, 2006) and each session focused on a set theme. The sessions included psychoeducation, exercises, and distribution and follow-up of homework. A recurrent feature throughout the treatment was exercises aimed at reducing the urge to gamble through imaginary exposure and response prevention. The treatment was focused partly on cognitive restructuring and partly on encouraging clients to try alternative behavioral strategies. Another important treatment component dealt with identifying personal high-risk situations for gambling and increasing participants’ skills to cope with these situations in a more functional way.

The MI treatment (n = 28) was administered individually in four 50-minute sessions. The first two sessions were one week apart, and the last two sessions followed at three-week intervals, for a total treatment time of 8 weeks—the same as the CBGT condition. The therapists used the MI approach as described by Miller & Rollick (2002), including showing empathy, eliciting the participant’s own reasons for making a change, collaborating with and supporting the participant in autonomy, developing the discrepancy between ongoing problematic behaviors and the participant’s internal goals and values, and supporting the participant’s confidence in their own abilities. Techniques such as open-ended questions and reflective listening were used throughout the sessions. If the patients were ready to change, they were encouraged to make a decision about changing their gambling behavior and to make a change plan. The therapists had access to a semi-structured manual in which these standard MI principles were described and exemplified in the context of problem gambling (Forsberg, Forsberg & Knifström, 2010).

Treatment fidelity

The therapists administering the CBGT received continuous supervision. All sessions were audio-taped and 20% were randomly selected for coding by an independent licensed clinical psychologist with psychotherapist training and experience in the specific treatment method. According to the treatment manual (Ortiz, 2006) a total of 375 agenda points should be covered. The coding showed 93% adherence to the manual.

To test MI treatment integrity, all sessions were audio-taped and 20% of the sessions were randomly selected to be coded by independent and blinded coders using the Motivational Interviewing Treatment Integrity Code 2.0 (MITI; Moyers et al., 2003). The MI competency in the delivered sessions was deemed acceptable using the given reference values for MI proficiency in the coding manual (Moyers et al., 2003). Supervision of the MI treatment was accomplished through assessment of the therapists’ audio-taped sessions. Results from the coding were used to facilitate specific feedback.

Statistical analyses

Analyses were conducted using SPSS 22.0 and STATA 14.0. Independent t-tests and Chi-square tests were used to determine whether the two treatment groups differed in pre-treatment characteristics. The same test statistics were also used to investigate whether participants who were lost at follow-up (n = 25) differed in pre-treatment characteristics from participants who completed the follow-up measurements. Analysis of covariance (ANCOVA) of the NODS scores and number of gambling days in the last 30 days at the 6-month follow-up was used, with the NODS scores and number of gambling days in the last 30 days at pre-treatment used as the models’ quantitative control variables. The final two models (one for NODS scores and the other for number of gambling days) had two factors, treatment (MI vs. CBGT) and AUDIT (risky vs. not risky alcohol habits), with an interaction term included in the models. Marginal means were calculated from the ANCOVA model and visualized (for the NODS scores) via a bar-plot of the margins. To assess the difference in NODS scores between the CBGT and the MI treatment adjusted for alcohol habits, contrasts of discrete marginal effects were estimated and tested.

Results

Pre-treatment variables

Table 1 shows baseline point estimates and the distribution of some basic characteristics of the participants (n = 53). No statistically significant differences in characteristics were found between the two treatment groups.

Table 1 Participants’ characteristics at pre-treatment including 95% confidence interval (CI 95%).

	CBGT (n = 25)	MI (n = 28)		
Characteristics	Mean	CI 95%	Mean	CI 95%	p-value	
NODS No risky alcohol habits	6.1	5.1–7.2	6.0	5.1–6.9	.84	
NODS Risky alcohol habits	6.2	5.3–7.1	5.7	5.1–6.3	.33	
AUDIT No risky alcohol habits	2.2	.9–3.5	3.6	2.3–4.9	.11	
AUDIT Risky alcohol habits	15.7	12.3–19.1	16.1	8.8–23.4	.92	
AUDIT-C	4.1	2.7–5.5	4.1	2.9–5.2	.99	
BDI	25.8	20.0–31.7	25.6	20.7–30.5	.95	
BAI	18.8	13.4–24.2	18.0	14.0–21.9	.81	
Age	43.0	37.5–48.4	40.8	35.9–45.6	.53	
Gambling debt: 1,000 USD	10.2	5.2–15.3	8.7	4.0–13.4	.65	
	Proportion	Proportion	p-value	
Female	20.0	17.9	.84	
Prior gambling treatment	40.0	46.4	.64	
Prior psychiatric treatment	44.0	57.1	.34	
Only elementary school	32.0	32.1	.99	
Immigrant	24.0	39.3	.23	
Unemployed	16.0	14.3	.86	
Low income	16.0	21.4	.61	
Primary gambling on:				
Video lottery terminals	56.0	46.4	.49	
Horse/sport betting	16.0	25.0	.42	
Casino/poker	12.0	10.7	.88	
Other	16.0	17.9	.86	

Figure 1 Marginal means and standard errors for interaction effects between treatment and alcohol habits.

NODS scores at 6-month follow-up

The interaction between treatment and alcohol habits in the ANCOVA-model was significant [F (1, 48) = 5.39; p = .025], and suggests that the effect of treatment depends on the patient’s alcohol habits. Although none of the factors in Table 1 differed significantly between treatment groups, we adjusted the model for gender, age, minority status, income level, gambling debts, and treatment attendance and found that none of these variables markedly changed the main estimates. Therefore, only the unadjusted estimates are presented. Marginal means calculated from the ANCOVA model showed that patients with gambling disorder and risky alcohol habits who received MI treatment had a mean NODS score of 1.9 at the 6-month follow-up. As the low average NODS score suggests, a strikingly large proportion (81.8%) of the participants in this group no longer met the criteria for gambling disorder at the 6-month follow-up. For patients with gambling disorder and risky alcohol habits who received CBGT, the corresponding NODS score was 4.0, with a lower proportion of participants (30.0%) who no longer met the criteria for gambling disorder at follow-up. The contrasts between MI and CBGT, shown in Fig. 1, were significantly different between participants with no risky alcohol habits and participants with risky alcohol habits [t(48) = 2.32; p = .025].

To confirm the results for the NODS scores, we used the same ANCOVA model with the number of gambling days per month as outcome measure. Unfortunately, there were only 39 observations for this outcome measure compared with 53 observations for the NODS scores, which might explain why no results from these analyses became significant. However, among participants in Zone I (alcohol habits), the means and standard deviations were 11.34 (1.87) for the MI treated and 8.86 (2.73) for those treated by CBGT. Among participants in Zones II–IV, the mean and standard deviations were 5.14 (3.45) for the MI treated and 11.18 (2.57) for those treated by CBGT. In other words, the results for number of gambling days were in line with the result for the NODS scores. These results suggest a better treatment outcome for CBGT than for MI among participants in Zone I and a better treatment outcome for MI than for CBGT among participants in Zones II–IV. In the first analysis that was performed, the AUDIT scores were analyzed with Zones II–IV as three separate categories compared with Zone I (reference). The risk estimate for Zone III versus Zone I was higher than the risk estimate for Zone II versus Zone I.

Analyses of missing data

There were 25 patients who did not participate in the 6-month follow-up, equally distributed between the two treatment groups, CBGT (n = 13) and MI (n = 12). There were no statistically significant differences (no p-values lower than .40) in terms of sex, age, or pretreatment scores on NODS and AUDIT between those who participated in the 6-month follow-up and those who did not.

Discussion

The findings in this study suggest that patients with gambling disorder respond differently to CBGT and MI depending on whether or not they have risky alcohol habits at pre-treatment. Patients with gambling disorder who also have risky alcohol habits appear to have a better chance of benefitting from MI, and patients with no risky alcohol habits appear to have a better chance of benefitting from CBGT. The results are clinically relevant because they can be used to facilitate the referral of patients with gambling disorder to the treatment that will help them the best. These findings raise the question of why MI appears to be more efficient than CBGT in treating patients with gambling disorder and risky alcohol habits, and why CBGT appears to be more efficient when patients do not have risky alcohol habits.

In a recent study on the personality traits of people with gambling problems with and without alcohol dependence, individuals with gambling disorder and co-occurring lifetime alcohol dependence reported a personality style characterized by resistance to externally motivated treatment approaches (Lister, Milosevic & Ledgerwood, 2015). In MI, patient behaviors characterized by resistance have been a focus of treatment, and such resistant behavior might be better addressed by MI than CBGT treatment. MI is a non-authoritarian, collaborative method that focuses on building intrinsic motivation (Miller & Rollick, 2013). The individually administered MI also offers more opportunities to tailor treatment to patient needs. MI is a compassionate treatment during which the patient is likely to feel comfortable raising personal issues (Miller & Rollick, 2013) that may pose obstacles to treatment if they are not given space. Risky alcohol habits could be addressed in MI treatment if it would help the patient to move towards the change goal, to stop or reduce gambling. The advantage of being able to address multiple behavior targets in MI treatment may have had a significant impact on the outcome because the two addictive behaviors are likely to trigger, reinforce, and maintain each other. Alcohol is usually readily available at casinos, racetracks. and other gambling environments, and gambling under the influence of alcohol is associated with higher risk-taking (Cronce & Corbin, 2010; Ledgerwood et al., 2009). Conversely, events that occur during gambling (e.g., winning and losing) may trigger alcohol consumption (Zack et al., 2005). Multiple behavior targets in MI treatment have been studied in other fields of addiction, and have proven to be effective in motivating people to simultaneously reduce their usage of tobacco, alcohol, and cannabis (McCambridge & Strang, 2004). In a review on smoking cessation during substance abuse treatment, Baca & Yahne (2009) concluded that targeting smoking cessation enhances outcome success and reduces substance use.

An additional advantage of the MI treatment over CGBT is that risky alcohol habits might have the same origins as the gambling disorder (Stewart et al., 2008). The MI therapist is therefore able to address risky alcohol habits in the treatment of gambling disorder, and the reasons for both alcohol consumption and gambling could then be highlighted and tackled from different angles.

In the CBGT treatment, on the other hand, the possibility of tailoring treatment to fit any comorbid conditions is very small because the treatment is in a group format and strictly follows a manual (Ortiz, 2006). The superior effect of CBGT on patients who did not have risky alcohol habits can probably be explained by the fact that CBGT was an extensive treatment that included a wide range of psychoeducative elements, exercises, and homework assignments that all addressed various aspects of problem gambling (Ortiz, 2006).

Strengths and limitations of the study

The major strength of this study is that it addresses the important issue of moderators of treatment effects. It highlights a factor that is highly correlated to gambling disorder and appears to moderate the outcome of treatment. The two treatment arms compared were evidence-based effective treatment methods for gambling problems, and treatment outcome was measured six months post-treatment, which implies that the results were persistent. The potential moderator (risky alcohol habits) included in the analysis was selected for two main reasons. First, it is the most common comorbid condition among people with gambling disorder (Bischof et al., 2013; Petry, Stinson & Grant, 2005); second, earlier findings indicate that the condition is an aggravating factor in treatment that correlates with impaired adherence to treatment (Milton et al., 2002; Rash, Weinstock & Petry, 2011), and increased risk of gambling relapse (Hodgins & El-Guebaly, 2010).

A limitation of this study is the small sample size, which made it difficult to include additional potential moderating variables in the model that would have been interesting to analyze. Additional potentially predictive comorbid conditions, such as drug use, mood, anxiety, and personality disorders should be included in future research. Unfortunately, a large number of patients dropped out at follow-up and were excluded from the analysis. This makes it necessary to be cautious in interpreting the results. The modified version of the NODS (assessing gambling at one month instead of one year) has not been evaluated. However, shortening the window of time from one year to 3 months does not appear to affect the instrument’s reliability or validity (Wulfert et al., 2005). Moreover, an apparent benefit of a shorter-term version of the NODS is that it can serve as a convenient treatment outcome measure. Another limitation is that it is unclear to what extent these results can be explained by different modes of treatment (individual vs. group) and to what extent they can be explained by unique factors inherent in each treatment. A final limitation is that there was no control group. It is therefore unknown whether the participants’ reported reductions of symptoms of gambling disorder during the 6-month post treatment period were the results of the treatment or spontaneous recovery. About one-third of individuals with gambling problems are believed to recover without formal treatment (Slutske, 2006). However, we have no reason to believe that the rate of spontaneous recovery should be different between treatment groups.

Generalizability

There were missing data at the 6-month follow-up. However, there were no statistically significant differences between those who participated and those who did not participate in the follow-up in terms of sex, age, severity of problem gambling, and alcohol problems at baseline. It appears reasonable to conclude that it is possible to generalize the findings to gamblers seeking treatment for gambling problems serious enough to meet the criteria for gambling disorder. The findings are interesting from a health-planning perspective, and are valid for both CBGT and MI, which are two commonly used evidence-based treatments for gambling disorder (Gooding & Tarrier, 2009; Hodgins, Stea & Grant, 2011; Yakovenko et al., 2015).

Future research

First, the results from this study need to be replicated to ensure that these associations are not sample-specific. In order to confirm our results, future studies should state a priori the hypothesis that people with gambling disorder and risky alcohol habits will benefit more from MI than from CBGT, and that people with gambling disorder but no risky alcohol habits will be helped more by CBGT than by MI. Further research would improve the validity of the findings if an intention-to-treat analysis were conducted. Moreover, further research is needed to investigate how other comorbid conditions, such as depression and anxiety, affect the efficacy of treatment. It would also be useful to learn more about the impact of comorbidity on individual CBT.

Conclusions

The results support a screening procedure including the AUDIT prior to the start of treatment for gambling disorder because the result of the screening can provide guidance in the choice of treatment. Patients with gambling disorder and risky alcohol habits are more likely to be helped if they are referred to MI treatment, while those without risky alcohol habits are likely to be best helped if they are referred to CBGT.

Supplemental Information

Data S1 Raw data

Click here for additional data file.

The authors would like to thank Sarah Heinrich for her valuable comments made to a previous version of this manuscript.

Additional Information and Declarations

Competing Interests

Author Contributions

Human Ethics

Data Availability

The authors declare there are no competing interests.

Henrik Josephson and Ingvar Rosendahl analyzed the data, wrote the paper, prepared figures and/or tables, reviewed drafts of the paper.

Per Carlbring and Lars Forsberg wrote the paper, reviewed drafts of the paper.

The following information was supplied relating to ethical approvals (i.e., approving body and any reference numbers):

The study was approved by the regional ethics committee in Stockholm (2005/5:5).

The following information was supplied regarding data availability:

Raw data file has been uploaded as Supplemental Information.

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
