# Peer review of "People with gambling disorder and risky alcohol habits benefit more from motivational interviewing than from cognitive behavioral group therapy"

_PeerJ, doi:10.7717/peerj.1899_

## Round 0.1 · original submission · Major Revisions

· Academic Editor

Major Revisions

Please respond to the reviewers concerns, with particular attention to a reworking of your analyses as suggested by reviewer 2. Both reviewers thought this was potentially publishable, but indicate that additional work is required to improve the paper,

·

Basic reporting

My concern here is that the results of the RCT this secondary analysis of data is using, are not reported and it would have been helpful to have understood how both treatment groups compared to each other and to the waiting list control. Adding this to the introduction would further clarify the impact on analysing according to hazardous drinking or not. Were both treatments equally effective without differentiating according to drinking?
Secondly, I personally find the term “Disordered Gamblers” unnecessarily clustering, demeaning and possibly inaccurate (the participants are not gamblers who are “disordered” as opposed to ordered dependent gamblers). DSM V refers to Gambling Disorders and people who have gambling disorders as problem gamblers (or people with problematic gambling behaviors). The report should avoid clustering language and I suggest replacing with DSM V terms.

Experimental design

As the authors do state, there is a small N with a large drop out at follow-up. It would be useful to know what the power calculation was for the original RCT and to use conservative terms for describing the results such as results “suggest” rather than “indicate” and add some cautionary words about the small N and statistical significance requiring some measure of clinical significance of these differences found.
On Line 229, the authors state a hypothesis that the research tested but it appears different from the hypothesis given in the introduction. The research does not empirically test “the common belief that comorbidity indiscriminately affects all forms of treatments negatively”, it tests whether screening for risky alcohol habits can provide guidance on whether disordered gamblers should be recommended cognitive behavioural group therapy or MI. This should be amended to reflect the latter hypothesis.
In the discussion and elsewhere, it should be clarified that the drinker’s group were people scoring above the hazardous cut-off score and this does not necessarily mean they are harmful and/or dependent drinkers. So, the results should be described as such and a limitation added that this study cannot show if severity of problem drinking impacts the effect found (it could for example be the case that MI differentially helped (i.e. reduced NODS scores) less severe problem drinkers more than dependent drinkers).

Validity of the findings

Please see my comments in previous categories that are also relevant here.

Additional comments

In general this is a good and well executed piece of research which undoubtedly adds to our understanding of problem gambling treatment. I would have appreciated a little more description of the original research findings which would have helped contextualise the present findings. There is a lack of critical analysis of the literature which tends towards being largely descriptive only. Some of the studies reviewed are limited in their applicability to the current one, and this could have been critiqued further.

Reviewer 2 ·

Basic reporting

In terms of 'unit of publication', I believe more could and should be done to raise the level of impact of this paper. The topic is important and timely and will likely move the field forward in terms of alcohol/gambling treatment research and it will inform clinical decisions. However, the analyses are narrow and lack thoroughness. The manuscript would also benefit from a thorough proof-reading for typos and misspellings.

Experimental design

Table 1 needs to be expanded considerably to better compare the groups on basic demographic, clinical, gambling, and alcohol characteristics. How much/how often do these patients drink? What types of gambling are prevalent? What's the average gambling debt? Reasons for seeking treatment? Number of prior treatments? Minority status? Income level? Percent employed? Subsequent analyses will need to be adjusted to accommodate any significant between group differences.

Validity of the findings

Conclusions are appropriately stated; however, the analyses are simplistic and the data reported lacks detail. For example, could the authors look at whether predictors of time to relapse differ between the risky drinkers and non-drinkers? What about parallel analyses looking at gambling frequency/expenditure? What about the impact of treatment attendance - does treatment attendance mediate the observed effect (which would lend strength to the authors conclusions)?

Additional comments

I noticed a discrepant sample size in the abstract versus what was used for analyses. The abstract reports 80 subjects, but all analyses were conducted on 53 subjects with full data. The abstract should be corrected.

This is a great topic and I would like to see the authors develop the analyses more thoroughly.

---

## Round 0.2 · Minor Revisions

· Academic Editor

Minor Revisions

Thank you for the revised version of the paper. Please consider the final suggestions of the reviewer regarding the discussion section and ensure that you have the final version thoroughly proof read.

·

Basic reporting

The authors have appropriately addressed the comments made on the first submission by me. There remains a large number of careless typos which need to be attended to.

Experimental design

No comments

Validity of the findings

The authors have addressed the comments well. I believe a further statement should be added in the discussion that as a large number of participants dropped out at follow-up and were excluded from the analysis, this is a criticism of the study. In addition state that further research would improve the validity of the findings if an intention to treat analysis were conducted.

Additional comments

I believe this submission is of a high enough standard to be published. It will be an important addition to the scientific literature and is of relevance to clinical practice.

---

## Round 0.3 · accepted · Accept

· Academic Editor

Accept

Thank you for addressing the concerns and engaging a proof reader.